# Emerging technologies and research ethics: Developing editorial policy using a scoping review and reference panel

Simon Knight[1,2,3]*, Olga Viberg[2], Manolis Mavrikis[3], Vitomir Kovanović[4], Hassan Khosravi[5], Rebecca Ferguson[6], Linda Corrin[7], Kate Thompson[8‡], Louis Major[9‡], Jason Lodge[5‡], Sara Hennessy[10‡], Mutlu Cukurova[3‡]

1 Centre for Research on Education in a Digital Society and TD School, University of Technology Sydney, Sydney, Australia, 2 Digital Futures, KTH, Stockholm, Sweden, 3 UCL Knowledge Lab, University College London, London, United Kingdom, 4 Learning Futures, University of South Australia, Adelaide, Australia, 5 Institute for Teaching and Learning Innovation / School of Education, University of Queensland, Brisbane, Australia, 6 Institute of Educational Technology, The Open University, Milton Keynes, United Kingdom, 7 Deakin Learning Futures, Deakin University, Melbourne, Australia, 8 Faculty of Creative Industries, Education & Social Justice, Queensland University of Technology, Brisbane, Australia, 9 Manchester Institute of Education, The University of Manchester, Manchester, United Kingdom, 10 Faculty of Education, University of Cambridge, Cambridge, United Kingdom

☉ These authors contributed equally to this work.
‡ KT, LM, JL, SH and MC also contributed equally to this work.
* Simon.Knight@uts.edu.au

**Data Availability Statement:** Relevant data are contained within the manuscript, its Supporting information files, and the linked data publication https://doi.org/10.6084/m9.figshare.26013130.v2;

## Abstract

### Background

Emerging technologies and societal changes create new ethical concerns and greater need for cross-disciplinary and cross–stakeholder communication on navigating ethics in research. Scholarly articles are the primary mode of communication for researchers, however there are concerns regarding the expression of research ethics in these outputs. If not in these outputs, where should researchers and stakeholders learn about the ethical considerations of research?

### Objectives

Drawing on a scoping review, analysis of policy in a specific disciplinary context (learning and technology), and reference group discussion, we address concerns regarding research ethics, in research involving emerging technologies through developing novel policy that aims to foster learning through the expression of ethical concepts in research.

### Approach

This paper develops new editorial policy for expression of research ethics in scholarly outputs across disciplines. These guidelines, aimed at authors, reviewers, and editors, are underpinned by:

1. a cross-disciplinary scoping review of existing policy and adherence to these policies;

where data cannot be shared publicly due to copyright restrictions links to retrieve the original source data are provided.

**Funding:** The author(s) received no specific funding for this work.

**Competing interests:** The research reported was instigated as a collaboration between the Journal of Learning Analytics (JLA), Australasian Journal of Educational Technology (AJET), and the British Journal of Educational Technology (BJET), by the lead author, who was a previous co-editor-in-chief of the JLA. The authors acknowledge BJET co-editor-in-chief Cathy Lewin, who joined the journal later in this process, and thus did not participate in authoring this piece. The views expressed are those of the authors, and may not represent the views of the journals, scholarly societies, or other organisations to which they are affiliated. The work was begun over the course of the lead author's sabbatical, which included periods co-located at UCL and KTH / the Swedish Digital Futures research centre, supporting direct interaction with editors based at those institutions. During the write up of this research, the lead author received funding from the Australian Government through the Australian Research Council (ARC) Discovery Early Career Award (DECRA) Fellowship (DE230100065), and Discovery Project (DP, DP240100602). The views expressed herein are those of the authors and are not necessarily those of the Australian Government or Australian Research Council. This does not alter our adherence to PLOS ONE policies on sharing data and materials.

2. a review of emerging policies, and policies in a specific discipline (learning and technology); and,

3. a collective drafting process undertaken by a reference group of journal editors (the authors of this paper).

## Results

Analysis arising from the scoping review indicates gaps in policy across a wide range of journals (54% have no statement regarding reporting of research ethics), and adherence (51% of papers reviewed did not refer to ethics considerations). Analysis of emerging and discipline-specific policies highlights gaps.

## Conclusion

Our collective policy development process develops novel materials suitable for cross-disciplinary transfer, to address specific issues of research involving AI, and broader challenges of emerging technologies.

## 1. Introduction

### 1.1. Learning to meet ethical challenges in a changing world

Across disciplines, researchers face challenges in navigating ethical issues regarding emerging technologies and changing societal context. The dual challenge is that existing strategies for applying an ethical approach to achieving positive impact in research may not align well with emerging topics, and that there may not be clear consensus or established cross-disciplinary resources to support understanding and navigating the ethical dimensions of such work. As the primary medium of research communication, scholarly outputs serve as a safeguard on research ethics 'compliance', alongside providing an opportunity for learning regarding navigation of existing and emerging ethical challenges.

Recently, a significant body of regulatory and other guidance has emerged in response to high-profile cases that raised ethical concerns regarding the much-vaunted potential of artificial intelligence (AI) in the public sphere. These challenges are made more pressing by the rise of AI, including the latest developments in generative AI technologies and technologies that may have dual use applications [1]. Four reviews of contributions to AI principle and guideline development report, 36 [2], 84 [3], 112 [4], and 27 [5] documents respectively. However, it is unclear how scholarly publishers and research institutions should incorporate these guidelines.

In this context of AI's growth and corresponding heightened awareness of the ethical challenges of emerging technologies, a number of learned societies and publishers are reviewing their role in promoting ethical practice within research, and positive impacts from research [e.g., 6–8]. Prominently, NeurIPS–a respected AI, machine learning, and computational neuroscience conference–introduced an 'impact statement' in 2020, switching to a checklist and review in 2021 [9]. In their recent report on the role of conferences in fostering ethics, "A Culture of Ethical AI", the Ada Lovelace Institute highlights the need for researcher incentives

regarding ethical considerations of AI research, which permeate the research development to dissemination cycle [7].

Editors (including conference program chairs) play an important role in promotion of ethical practice because publication is a key aim for many researchers and serves as an instrumental aim towards career advancement. If editors incentivise consideration of ethical issues, and mitigation of risks, this is likely to impact research conduct and expression [1, 7]. However, despite longstanding relatively procedural requirements regarding ethics, and specifically the reporting of ethical-oversight approvals and consent processes, understanding the expression of the ethics constructs and values that researchers work with is a challenge across disciplines [10]. A recent review of articles that discuss ethical issues in research (i.e., where this is a central theme), highlights that descriptive ethics dominates, with a relative lack of clarity regarding the concepts used or reflection on these [10]. Publications have a role in disseminating knowledge, including regarding the ethical considerations of our research, and ethics committees expect researchers to turn to the disciplinary literature in aligning their practice with disciplinary norms of ethics. Thus, this gap presents a challenge for, "increasing sensitivity to ethical issues and how empirical data may be relevant to various ethical principles and problems." [11, p.16].

The present paper bridges this gap through systematic analysis of existing editorial policy, and development of new policies targeting development of ethical sensitivity. While we draw on broad disciplinary context via a scoping review (not limited by field), we recognise that the inherent diversity within different scholarly domains requires tailoring to specific areas. Therefore, this paper provides exemplification through the area of 'learning and technology', targeting emerging issues regarding research ethics and AI applications in education. Our approach thus draws out general principles for editorial policies regarding the expression of, and learning about, research ethics, while acknowledging the inherent diversity across disciplines, and their distinct challenges.

## 1.2. Background: Reporting research ethics

**1.2.1. Recognising research ethics committees.** In the inception, conduct, and dissemination of research we navigate an ethics ecosystem [12, 13] that includes ethics guidelines and principles, alongside publication processes such as editorial policy that targets research ethics, and formal bodies for human and animal research oversight. These oversight bodies include research ethics committees (RECs), or boards (REBs), and institutional review boards (IRBs). RECs play a significant role in research ethics internationally, through their role in providing "ethical approval for study"–to use the refrain commonly seen in journal articles. RECs play this role in research governance, providing oversight of research conducted against ethical standards that typically incorporate the Belmont principles [14], alongside providing feedback to researchers against these values.

While this oversight is grounded in some common values and concern regarding historic ethical abuses, it is important to recognise that: 'ethical approval' from a REC may not indicate the research is ethical; RECs are sometimes perceived to hinder ethical research [15]; ethics is bound up with contextual power relationships and historical injustices [16]; and that the scope of REC reviews is limited, for example, in the US, IRBs are explicitly instructed not to consider long-range effects that would include the risks of technologies should they be deployed at scale [for discussion, see 17].

Of significance for editors, ethics guidance and governance also varies internationally, and it may not be necessary (or even possible) for researchers in some systems to gain formal approval for research projects, with the nature of review structures, and of what counts as

'human research', varying internationally [18–24]. Editors should thus exercise caution in imposing requirements or scoping the nature of 'research', in ways that could unduly exclude research, and for example, entrench existing inequities, marginalise voices in research, or exclude methodological traditions.

**1.2.3. Reviews of research ethics publication policies.** Given this important role for editors in research governance, international guidance indicates that scholarly venues should require processes of consent and ethical approval as part of submission processes. Existing policies focus on the explicit expression of key ethical principles in publications, often including discussion of principles connected to the medical research Declaration of Helsinki [25] or Belmont Principles or Common Rule [14]. While they serve different purposes, both include statements relating to respect for persons, beneficence, and justice, with operationalisation into publishing often focusing on explicit statements regarding informed consent, and oversight or approval of a REC as expressed in recommendations of the International Committee of Medical Journal Editors [26] and the Uniform Requirements for Manuscripts Submitted to Biomedical Journals (URMs) [27].

While historically these statements arose from medical research contexts, the Committee on Publication Ethics (COPE) and World Conferences on Research Integrity (WCRI) have developed more general material. The WCRI's Singapore Statement final item flags the significance of "societal considerations" and a balance of risks and benefits in research, and COPE's "Responsible research publication: international standards" for authors and editors, noting appropriate REC approval and its reporting, consent, and privacy as key issues, alongside wider concerns for research integrity and merit [28, 29]. As COPE's self-assessment for editors sets out, venues: "must publish clear guidelines on the ethical conduct of research, according to the research discipline" [30] as part of their policies for COPE Core Practices. However, despite COPE's multidisciplinary status, their survey of 656 editors in the arts, humanities, and social sciences indicated that "28% of respondents were completely unaware of COPE" [31, p.3]. Even in medical publishing, which has a longer association with the Declaration of Helsinki, a survey of editors of journals (n = 34 respondents) indicates not all give explicit instruction to reviewers regarding ethics (38% do not), and some (18%) report that they had published 'ethically uncertain' or suspect' research previously [32].

Several reviews have been conducted regarding journal policies for research ethics, largely instantiated through their 'instructions to authors' (ItAs) and alignment with the Declaration of Helsinki and International Medical Research statements (requiring ethics approval, and consent processes) (see Section 2.1). Although these reviews have been executed across disciplines, there is no synthesis across them which might aid in identifying areas of strength or/ and strategies being adopted by different disciplines or publishers, particularly in light of addressing challenges posed by emerging technologies.

## 1.3. Editorial policy to support learning regarding research ethics

Editors play a role in a wider ecosystem of research ethics and impact [12, 13]. As Gold et al. put it, "publishing communities (e.g., scholarly conferences) can play a larger role in supporting improved ethical practice by defining and communicating the ethical values of their community's collective identity and aspirations" [33, p.1]. Recent reports regarding the challenges of AI as an emerging technology have set out focal areas for editors in fostering ethical practice in their research communities [7, 34, 35]. To address these, editors have available to them a range of policy levers, articulated into material resources for their communities, that express standards and models for change, as Table 1 summarises. Notably, while Table 1 summarises

**Table 1. Foci and policy levers for fostering ethics in research publications.**

| # | Strategy focus | Policy lever |
|---|---|---|
| 1 | (1) Prescriptive and reflexive interventions to foster ethical reflection*; | **Instructions to:** authors, reviewers, and guest-editors/editorial-boards regarding the requirement to include particular issues |
|   | e.g. "disclose and report additional information in their papers"^; "review potential downstream consequences earlier in the research pipeline"^ | **Submission template requirements** (published within article). This might include supplementary or structured elements, such as notes for practice (already adopted in some venues), or drawing on the range of resources available such as model cards or canvases of various kinds etc. |
|   | "expand peer review criteria to include engagement with potential downstream consequences and establish separate review processes to evaluate papers based on risk and downstream consequences."^ | **Submission *form* requirements** (not published, or published as metadata). These include checkbox confirmations and fields included as part of the submission process. |
| 2 | training for reviewers and researchers*; | **Training provision** alongside instructions to authors, reviewers, etc. This might include: mentoring or author support particularly for junior authors; worked 'cases', examples of practice, or resources to benchmark author statements against; workshops/tutorials e.g., at conferences, covering aspects of the research-publication ecosystem and ethics, etc. |
| 3 | engagement with stakeholders impacted by tools*; | **Soft policy** to encourage research that engages with stakeholders in the design, development, evaluation, and implementation of tools. (see 1) |
|   |   | **Space within venues** for reflection on implementation or ethical engagement (see 4) |
| 4 | specifically drawing attention to work that exemplifies technical and ethical principles*; | **Submission *categories* or types**, to spotlight or provide space for particular types of discussion within articles, or/and dialogue among articles. |
|   | "commend researchers who identify negative downstream consequences^;" | **Awards** or other spotlighting mechanisms to highlight key work. |
| 5 | incentivise 'slower' research to support e.g., rolling submissions and R&R in conferences (rather than a one-shot speedy output)*; | **Policies to foster research**: Replicability (e.g. code/data sharing), Replications, and Replies (e.g., post-publication-review, commentaries re: implementation in practice, etc.) |
| 6 | Space for dialogue regarding the normative concerns of research. (1) more research on the effects of ethics review processes+; (2) more experimentation with such processes themselves+; (3) the creation of venues in which diverse voices both within and beyond the AI or ML community can share insights and foster norms+; | **Venues for dialogue:** 1. Materials to actively promote discussion including via editorials 2. Hosting workshops, round-table discussions, etc. on the topic 3. Provision of worked 'cases' |
|   | "normalize discussion about the downstream consequences of research"^ | 4. Resources to support consideration of both participant, and societal impacts over time |

Sources indicated as:

*[7]; +[35, p.1061];

^[34, p.1].

proposals targeting issues arising from emerging technologies, most have broader impact in developing practices for research ethics.

## 1.4. Research questions and impetus

Central to the policy levers expressed in Table 1 is the issue of how scholarly outputs express and support learning regarding the navigation of ethical implications of research, particularly in light of emerging technologies. It is imperative that we have suitable policies and practices across communities with this learning focus. Our claim is not that current research fails to consider ethical concepts, but rather, that greater attention would lead to deeper expression in order to (1) support learning within and across research communities, and (2) develop knowledge regarding the communication and development of these ethical concepts. However, the ways that publishers, editors, and authors, can support learning regarding ethics has not been a significant feature of research or policy statements to date. Thus, as Table 2 sets out, this paper provides an in-depth examination of the ways ethical practice is enacted and communicated in academic publishing so it can be incorporated into processes that enable editorial

**Table 2. Research questions, approaches, and outcomes.**

| n | Research Question | Approach | Outcome |
|---|---|---|---|
| RQ1 | *What is the scope of editorial policies addressing research ethics, and their instantiation into published works?* | A scoping review was first conducted to (a) identify editorial policies addressing research ethics, and (b) examine compliance with such policies. | An overview of the range of policies in use (a 'policy menu'), and the issues they sought to address (see §2.1) |
| RQ2 | *What policies are adopted in a target set of journals from the discipline of learning and technology?* | We then focused specifically on learning and technology journals, reviewing the policies in place in that discipline, and conducting an analysis of references to ethics within published works in a single year across a subset of journals (see §2.2) | Analysis provided an in-depth perspective in a specific context to support operationalisation (extending the policy menu). |
| RQ3 | *What editorial policies for fostering expression of research ethics are recommended by an editorial reference group?* | Finally, we drew on the 'policy menu' and corpus of published works in the specific discipline to support our work as a reference group to co-author draft policies | Production of draft policies that may be adapted across disciplinary contexts (see §2.3) |

teams to enhance shared understanding and knowledge development of the editorial teams themselves, alongside researchers and stakeholders, around key ethical concepts.

# 2. Materials and methods

## 2.1. Scoping review

**Rationale and objective:** A search strategy was adopted to identify the range of policies regarding research ethics, and their instantiation in published works. Scoping reviews provide a systematic approach to understanding the scope of coverage of issues, while not seeking to systematically evaluate 'effectiveness' (per a systematic review) [36–38]. This approach is not intended to be exhaustive, but rather to capture the scope of coverage through 'high level' analysis, allowing for the identification of clusters and gaps that can inform the focus of future activities [39]. A strength of the methodology is its ability to identify key features of a diverse body of research in a connected manner [40]. The scoping review aimed to address RQ1 in order to provide insight into the range (or menu) of policies being used across venues in different fields, to inform our understanding of these existing resources and their gaps (Table 2). A PRISMA Checklist for scoping reviews is provided (S5 File).

The scoping review was not pre-registered.

**2.1.1. Search strategy.** *2.1.1.1. Eligibility criteria overview.* Editorial materials are sometimes the object of research publications, and occasionally they are published in the form of statements or editorials in ways that are indexed. More often though editorial policy might be identified via grey literature or analysis of journal/conference materials (primarily their websites). Research ethics policies known to have addressed emerging technologies (e.g., from the Association of Internet Researchers (AoIR), material on wearable technologies, etc., see [41]) were considered for inclusion, however these did not typically include policies of relevance for inclusion as editorial policy. Materials were investigated for both discussion of policies, and author behaviour or the instantiation of those policies into scholarly outputs. Materials were reviewed by the lead author and included if they discussed research ethics.

*2.1.1.2. Search strategy overview.* A search strategy was adopted to combine cross-disciplinary scholarly indexes with searches of target venue websites, and discussion in grey literature.

*2.1.1.3. Search terms.* Terms and filters varied by source (see S6 File), Scopus terms focused on research articles in which terms relating to "instructions to authors" and ethics appeared in the Title, Abstract, or Key terms; Google Scholar searches were for terms relating to "editorial policy" and "ethics reporting"; COPE resources were searched particularly for terms relating to ethics, alongside further purposive sampling.

*2.1.1.4. Quality appraisal.* No quality assessment was made of sources retrieved, consistent with conventional scoping review practice [42].

**2.1.2. Venues / sources.**   Sources were identified (detailed in S1 Fig) through:

1. Searches of scholarly indexes (Scopus; Google Scholar),

2. Searching COPE materials,

3. Purposive sampling including through citation chasing from the Ada Lovelace report,

4. Sampling from a set of Target Journals (see 2.2).

**2.1.3. Data extraction.**   Items were selected from returned results, and purposive sampling of publishers and venues (1) known to have developed innovative approaches particularly with respect to emerging technologies and AI (e.g., the NeurIPS policies, addressed to the field of computational neuroscience, including AI applications), or (2) within the specific discipline, through analysis of journal policies. Retrieved materials were analysed by the lead author, with these materials subsequently used in reference group discussion. Analysis included both quantitative information (i.e., policy or compliance incidence), and descriptive information, particularly concerning the discussion of policy need or policy innovations including:

1. Discussion of editorial policies regarding research ethics. This included material reviewing editorial policies from journals in a specific discipline and their explicit mention of the requirement to report consent, as well as discussion of novel cases or introduction of new policies (particularly in the COPE and purposively sampled material).

2. Discussion of compliance with editorial policy regarding research ethics (e.g., reviews of papers in a specific discipline and their discussion of consent).

Items were imported into the Zotero reference management tool [43] for categorisation, connection, and data extraction, where:

1. They were 'Related' using the 'related' function, such that if an item referred to another, they were linked (e.g., policies linked to published works they referred to).

2. The unit of analysis was identified as content related to research ethics, ranging from a paragraph or subsection, to a single bullet point or checkpoint. For many publications, the abstract contained the relevant information.

3. Units were extracted where they referred to research ethics (correspondingly, item elements that referred to other aspects of research integrity and merit were excluded, except where they were clearly linked to discussion of research ethics).

4. Extracted units were coded by strategy focus and policy lever, using an excel export of the Zotero data (see data file [44]).

## 2.2. Target journal sampling

Learning and education research faces distinct challenges in consideration of the role of technology in research, and in learning environments. The ethical component of these challenges, alongside the potential for positive impact, has been long recognised [45], with recent calls for clear policy regarding inclusion of ethics statements in one leading AI in Education journal [46]. As experts in this discipline, we focused on the field to situate our policy development,

while providing a model for transfer across disciplines. To identify journals of interest the lead author:

1. Used the SJR rankings for 'e-learning' (n = 72 journals), and systematically visited the site of each in the first and second quartile (n = 36) for screening;

2. Screened venues for whether (1) they were associated with a society (n = 9), and (2) they had a focus on learning or/and the role of technology in learning (n = 7 of that 9);

3. Identified a journal representing each publisher (where multiple journals were identified with the same publisher convenience sampling was used based on the lead author's existing network);

4. Appended the list with purposive sampling of known journals (n = 6) in the space (adding two additional 'self-created' ethics statements).

This list was used to invite journals to participate (see 2.3), and for extraction of policies, with a review of pages representing all publishers for references to 'ethics' (9 generic boilerplate materials were identified from publishers: Wiley/ IGI/ Emerald/ PKP/ T&F/ Springer/ Inderscience/ IEEE/ informingscience). Further detail is provided in published data [44].

**2.2.1. Paper sampling within target journals.**   To understand how 'ethics' is incorporated into published works in the field, permission was sought from the subset of journals whose editors were on the reference group (n = 3, see below), to analyse their published works for the year 2021. Two searches were conducted with a narrower and broader query, with summary statistics and a Key Words in Context (KWIC) output created using an R script, in order to use these materials as stimulus in the expert consultation.

## 2.3. Reference group consultation

**2.3.1. Recruitment approach.**   A reference group of journal editors was invited to collaborate, with an aim to discuss the issues they encounter, and possible policy directions. Invites were sent January 26 2023 with a request to respond (with a decision, or request for more time) by February 7th; editors of n = 6 journals were invited, editors of n = 3 journals accepted. In discussion with those editors, it was decided to proceed without a further round of invites. A subset of these editors led consultation with their 'Editor-in-Chief' teams, with the full team contributing to policy development and this manuscript. The invitation made clear that insofar as policies might be an outcome, the intent was not to seek a universal mandate, but to develop adaptable material that would respect the plurality of values and the needs of communities being served.

Editors of the journals were selected on the following basis: (1) each journal focuses on learning and technologies; (2) there is some variation in editor location across the journals; (3) a range of publishers (e.g., Sage), societies, or/and publishing platforms (e.g., the open source PKP OJS platform) are represented each of which provides their own boilerplate policies for journals. There was a pre-existing relationship between the lead author and all invited editors. Of those invited, three journals responded positively and thus form the reference group (n = 6 editors comprising a single representative of two, and the full team of one journal, n = 1 convenor). This process is represented in an adapted PRISMA Flow, S7 File.

**2.3.2. Ethical considerations and consent.**   The editors were invited in the course of their ordinary duties, with an indication that the work would be collaborative and–subject to ICMJE authorship guidelines–involve co-authorship. The project was approved under the 'negligible risk' category for consensus methods (ETH23-7991 at UTS, ratified at UCL ExREC0005); participants were provided an information sheet outlining the implied consent

approach "Your participation in this work will be treated as confirmation that you have read this information sheet, and consent to participate." This was also noted at the start of the first meeting, alongside expectation setting and initial discussion of any ground rules regarding, for example, use of quotes from our meeting sessions.

**2.3.3. Participation approach.** The group met for four virtual workshops collectively over a 6-month period, with sub-group meetings and asynchronous collaboration targeting:

1. Co-developing and promoting an approach to drawing on expertise in learning, to review and develop practices and editorial policies with AI ethics as a shared initial impetus.

2. Contributing to evaluation of existing, and drafted, policies and development of policies aiming to produce materials to be adopted/adapted across disciplines.

## 3. Results

### 3.1. Scoping review across disciplines

The review identified resources as indicated in the modified PRISMA Flow diagram (S1 Fig, with further detail of the review process and search terms in S6 File). The subset of these resources identified in Google Scholar and Scopus searches that reported empirical reviews addressing ethics policies and reporting is indicated in Table 3.

These papers were reviewed to inform our understanding of the existing policy landscape and reflections on this for policy development. Where available, policy foci and materials were extracted from papers even where they were excluded from the numeric summary analyses provided.

**3.1.1. Summary of policy incidence.** On inspection of the papers focusing on policy, those focusing solely on ItA issues outside research ethics, such as conflict of interest statements, were excluded (n = 4); a number were not available (n = 3); some did not report data applicable for this review (n = 9); one further meta-analysis of systematic reviews of instructions to authors was identified (row 2 Table 3) [47], however that research did not report on the policy incidences identified in the papers it reviewed, and thus is excluded in further analysis, leaving n = 50. Analysis of those studies that investigated editorial policy regarding ethics guidelines indicates that across reviews of journals included (n = 4,440) many (n = 2,393 or 53.90%) do not include explicit statements regarding reporting of research ethics. These studies incorporate a range of journals, with varied disciplinary and geographic distribution, as does the number of venues included in any one analysis (one study analysed a single venue, M = 88.80, SD = 86.86, Median = 68). The average policy incidence in the journals included in each study was M = 47.86 (47.03%; SD = 60.11), i.e., almost half of journals appeared to have no explicit policy.

Where reference to specific external standards or guidelines was made, these were checked for inclusion in our 'policy menu', informing later phases. This checking of external resources is significant because in one review of articles (n = 324) analysing social media data, it was

**Table 3. Research ethics policy and reporting incidence in disciplines or journals.**

| *Discipline or journal incidence of ethics:* | |
|---|---|
| Reporting via outputs (n = 31) | Policies in venues (n = 64) |
| 14 (both) | |
| 17 | - |
| - | 50 |

found that, "*none of the five main publishing houses ha[d] any specific policies on the use of [social media] data in research. Rather, all referred to their affiliations with the Committee of Publishing Ethics (COPE), which on investigation, also had no specific policies.*" [12, p.335].

In addition, the included studies investigate a range of issues, and present specific policy suggestions. An example is that studies in the discipline of psychiatry that may not be unethical, but might nevertheless be controversial, should explicitly report these issues, and that reporting guidelines should be extended to foster such discussion [48] In the discipline of ecology it is suggested that formal guidance should be developed for authors and reviewers regarding the ethics of research in sensitive areas, and ethical practice in work with Indigenous peoples or on traditional lands [49]. In clinical research involving photographs that may be re-identifiable, it is suggested that standards should be established with policy enforcement for deidentification [50]. While each of these issues relates to particular challenges of the discipline, in each case there are cross-disciplinary lessons in making explicit these concerns. Relevant to our focus (section 3.2), one Spanish language review examined the policies of 214 JCR indexed education journals and found 11.2% had no mention of ethical issues including issues of integrity, and animal and human considerations–corresponding to our focus on research ethics–covered in 23% of journals [51].

**3.1.2. Summary of expression incidence particularly as related to emerging technologies.** Of the papers discussing the incidence of expression of ethical concerns in published works, data could be extracted from n = 26, with two exclusions additional to those described above, due to different methodological focus. Across this 26, a total of 10,198 documents were reported as analysed (M = 407.92, SD = 613.41, Median 158). Across these papers, n = 5,212 (51.11% of the total) omitted discussion of ethics, a finding consistent across the papers (M = 208.48, 44.54%; SD 285.57; Median = 85, 49.04%). Corresponding to journal policies, many papers do not address key ethics requirements, across a range of disciplines and document types.

From these papers, concerningly in consideration of challenges arising from emerging technologies, one analysis of health technologies research indicates that papers: (1) typically (53% or 120/227) did not refer to ethical principles, and (2) focused on immediate participant impacts and the intent of interventions, rather than longer term evaluation of effects when technologies are implemented [52]. Similarly concerning for any research involving unequal power relationships or diminished capacity to provide consent, in research with Alzheimer's patients–who may have impaired decision-making capacity–almost half of articles reviewed (n = 125 articles across 62 journals) did not mention participant involvement in consent processes [53]. In a further paper of relevance to emerging ethical issues, a recent analysis of 132 articles that used publicly available data for discourse-analysis, roughly a third did not discuss ethics [54]. Similarly, Badampudi et al., [55] reviewed software engineering journals to investigate how researchers report the ethical issues around consent, confidentiality, and anonymity, finding that roughly half discussed ethical issues, but very few (6/95) addressed all three issues. Indeed as noted above, in analysis of 324 articles analysing social media data, only 8% (n = 25) reported having sought ethics review, and 20% (n = 65) provided a justification for not having done so, with journals not only providing little policy direction in this regard but instead pointing to external resources that would provide no suitable guidance to address this issue of research ethics [12].

## 3.2. Targeted content analysis

Across the items reviewed, the most frequent focus was on the instantiation of external policy such as the Declaration of Helsinki or ICMJE guidance into instructions to authors, or/and

published works in the venues (see Table 1). Other policies mentioned were tangential to research ethics, including reporting guidelines relating to methodological quality, and integrity issues such as conflict of interest. Some works appear to highlight gaps in existing guidelines, but it is not clear if these have been addressed, nor how they might be consolidated to support learning across disciplines. To further assess the range of policies, non-empirical, and grey literature sources (including venue policies) were assessed for their alignment with the foci identified in Table 1.

The data in this section were derived from the scoping review searches, in addition to purposive sampling both of disciplines in which innovations targeting emerging technologies were known to have occurred, and across a set of journals/publishers (n = 11 sources) within our specific discipline.

**3.2.1. Policy review: Recent innovations.** Policies vary with respect to their intent to be identified or expressed explicitly in other works. For example, the NeurIPS policies make clear that part of the intent of introducing a prescriptive review checklist, is to foster reflection and support dialogue in the wider research community regarding the impacts of research. In this way, it explicitly intends to become part of the discourse regarding those impacts. Early work has analysed the impact of the NeurIPS changes [9]. Through an analysis of the impact statements the authors highlight that the reviewer statements on the ethics requirement (introduced simultaneously) flag a desire for authors to elaborate further on their REC process and mention explicitly both what guidelines were followed and how [56]. A recent scoping review of publications on ethics in the ACM Digital Library–a large full-text index of computing publications–discussed a range of considerations including studies that provide helpful guidance in navigating emerging ethical concerns [57].

In many cases, excellent discussion will exist regarding ethical practice that is internal to research communities, including reflected through published articles. However, this discourse may not be reflected in editorial policy pages, thus limiting its inclusion in this scoping work. Nevertheless, the focus of most policy examples is "prescriptive and reflexive interventions", largely via instructions to authors and correspondingly to reviewers. Typically, this focuses on a narrow set of criteria as set out in IJCME or/and COPE, with other examples provided in supplementary material (S8 File, and published data).

**3.2.2. Policy review: Target venues.** From each venue, references to 'ethics' were investigated in all guidance provided, including searches, and following links in menus or embedded in guidance to identify discussion of research ethics. These were grouped with respect to the common policy instruments identified, and the ethics issues addressed (often briefly). Most guidance was provided via general instructions to authors, or through resources linked from policy pages, with–from the materials available to us–relatively little coverage in submission templates or processes, or reviewer materials (see Table 4). Moreover, the focus of discussion was often out of scope, providing guidance on issues of integrity but not research ethics more broadly, or/and on only parts of consent, ethics oversight, and consideration of identifiable data, or the wider set of ethical concerns relevant to research. COPE guidance was mentioned in many venues, but as noted above, COPE does not provide specific guidance on research ethics.

## 3.3. Editorial reference group

The group worked with the materials identified via the scoping review and targeted content analysis (which included the policies from the journals represented, S8 File), and an analysis of the published works from those journals for the year 2021, for terms relating to 'ethics' used to create summary statistics and a Key Words in Context (KWIC) output. From this, of the 210

**Table 4. Overview of editorial policy instruments referring to research ethics.**

| | ItA | Submission template | Submission process checklist to submit | General guidance and linked resources | Reviewer materials |
|---|---|---|---|---|---|
| **Emerald** | - | - | - | C, R, I | - |
| **IEEE** | - | - | - | - | - |
| **Inderscience*** | C*, G | - | - | - | - |
| **Informing Science** | - | - | - | R | - |
| **PKP generic +** | - | - | - | G | - |
| **PKP venue 1** | R | - | R, G | - | - |
| **PKP venue 2** | - | - | - | E | - |
| **PKP venue 3** | R, | - | - | D | G |
| **Springer** | C, R, D, G | - | - | - | - |
| **T&F** | - | - | - | R, D | - |
| **Wiley** | R, C, D | R, C | - | R, C | - |

C = Consent, R = REC approval, I = Identifiable data, D = Disciplinary commentary, G = Generic "Ethical considerations should be noted", E = External link, e.g. to COPE given but without further detail. Publisher pages are typically provided 'boiler plate' to journals, and so were reviewed for target journals whose titles are omitted here.

*Indicates signed consents must be provided on submission pre-review, this would be unusual in most social science disciplines. +PKP provides the Open Journal System (OJS) to open access venues with boilerplate text, but is not a publisher.

outputs (excluding 12 editorials), 186 (83.75%) include any references to ethics (which might include ethics review board, etc.). Uses of these terms were provided to the editors to support discussion regarding how research ethics is reflected in their journals. Editors were also invited to consider suitability of current policy, and examples of research or/and research dissemination where they had seen good practice, or issues that policy development might address.

The reference group was invited to:

1. Discuss the distinctive characteristics of research involving AI particularly in the disciplinary context of education research, as a way of framing our initial discussions in light of a perceived growing impetus;

2. Identify the set of stakeholders to whom editorial policy might apply and the policy instruments that might be used to foster expression of research ethics;

3. Develop draft materials to this end.

These discussions indicated that much of the concern in the space of editorial policy regarding research ethics for AI could have broader implications, both in considering other emerging technologies, and in considering wider concerns regarding research ethics and its communication. There are distinctive features of AI research such as black-box models and their implications for autonomy and consent. However, policies that would foster expression of ethical approaches to such issues and their sharing for wider community learning would generally apply to a wider variety of research. This lens thus provided the framing for four key outputs developed, with material from the scoping review addressing 1–3 below collated into the draft of each document, and informing its development:

1. 'Joint Statement' (S1 File): This document set out key terms of reference, or a manifesto, for framing the role of editorial policy in research ethics. The overarching position of this statement can be summarised by two key claims: that editorial policy regarding research ethics should: (1) Foster ethical practice and expression within our research communities; (2)

Support learning about ethical practice and concepts across the communities that engage with us as researchers;

2. Guidance for Authors (S2 File): This document is grounded in a review of ItAs, and draws on these and the joint statement to frame author guidance with respect to learning for ethics;

3. Guidance for Reviewers (S3 File): This document reviewed all checklists and other available reviewer guidance regarding research ethics, to consolidate, and identify themes in these, with a framing statement noting the role of reviewers in fostering learning regarding research ethics.

4. Guidance for Editors (S4 File): This document draws on the entire research project, providing a summary for editors regarding their role in research ethics, and approaches they might take in evaluating their practice and developing new policy.

To ensure that the perspectives of both the reference group and their wider editorial teams were properly represented in the materials, the cohort were invited to co-author a joint piece (this paper) regarding the shared imperative and outputs. In doing so we aim to share these materials, to inform further dialogue regarding research ethics and its expression in scholarly publications.

## 4. Discussion

Emerging technologies and changing societal context give rise both to novel ethical concerns, and increased need for cross-disciplinary and -stakeholder communication regarding the navigation of ethics in research. Given the role that journal and conference papers play in communication to scholarly and stakeholder communities, looking to them for insight regarding navigation of these ethical issues seems reasonable. However, as analysis arising from our scoping review shows, there are significant gaps in policy across a wide range of journals (57.80% have no statement regarding reporting of research ethics), and adherence (48.95% of papers reviewed did not refer to ethics considerations). If not in these outputs, where should researchers and stakeholders learn about the ethical considerations of research?

Our analysis of policies, both arising from the scoping review and from the specific discipline of learning technologies, indicates a range of approaches adopted, however with significant variation in coverage of ethical concerns, and gaps, and a prescriptive policy focus largely on ethical oversight. We provide an overview of the state-of-the-discipline, highlighting key policies to support their uptake. We would also re-iterate a point made earlier in this paper, that many venues refer to COPE guidance, despite COPE providing limited specific guidance on research ethics.

Using this review, our reference group meetings noted that (1) many of the issues raised in light of AI are not specific to AI, and (2) many issues are longstanding. There are some specific expressions of ethical concern in research involving AI, including concerns regarding: consideration of long-range impacts; adequacy of oversight for research and its outputs in exploratory and developmental research stages (e.g., creating an algorithm) vs, in implementation and deployment stages (e.g., the algorithm being taken up in a product); and the secondary uses of data for unanticipated purposes. However, these concerns are not always expressed in frameworks, and highlighting their novel nature may help stakeholders in understanding how different research contexts give rise to particular ethical concern. To address gaps in existing policy, we highlighted the potential for a policy shift to emphasise the potential for scholarly outputs to support learning regarding the ethical issues encountered and their navigation.

### 4.1. Limitations

Our approach drew on a scoping review of research regarding research ethics and materials specifically targeting this issue drawn from journal and conference materials. The intent of this scoping review was to gain a view of the policies being used and adhered to, and how these might be drawn on in our reference group. This approach is limited insofar as the scoping review searches are not exhaustive (nor are they intended to be), the materials review was led by the first author, and the policy development is from a subset of editors, in a particular field. Most significantly, the leap from policy creation to implementation in venues, and assessment of the impact of such policies, is a challenge for future work.

### 4.2. Conclusion

This paper sought to learn from the existing policy landscape, to practically develop materials that might be adapted/adopted across disciplines. While our initial impetus was use of AI in education, our analysis indicated broader potential for addressing longstanding concerns, and thus the materials developed provide indications for where they may be modified for different disciplines and contexts, and have broad application in seeking to foster learning and constructive dialogue regarding the ethical implications of research and its conduct.

## Supporting information

**S1 Fig. Modified PRISMA flow diagram of scoping review search and process.**
(DOCX)

**S1 File. Joint statement on editorial policy to foster learning regarding research ethics.**
(DOCX)

**S2 File. Guidance for authors on their role in fostering learning regarding research ethics.**
(DOCX)

**S3 File. Guidance for reviewers on their role in fostering learning regarding research ethics.**
(DOCX)

**S4 File. Guidance for editors on their role in fostering learning regarding research ethics.**
(DOCX)

**S5 File. Preferred Reporting Items for Systematic reviews and Meta-Analyses extension for Scoping Reviews (PRISMA-ScR).**
(DOCX)

**S6 File. PRISMA-flow with detailed search information.** Modified PRISMA Flow Diagram of Scoping Review Search and Process.
(DOCX)

**S7 File. PRISMA-flow of journal invites.** Modified PRISMA Flow Diagram of Identification of Relevant Journals for Reference Group Invitation.
(DOCX)

**S8 File. 'Policy menu' examples.**
(DOCX)

## Acknowledgments

The research reported was instigated as a collaboration between the Journal of Learning Analytics (JLA), Australasian Journal of Educational Technology (AJET), and the British Journal of Educational Technology (BJET), by the lead author, who was a previous co-editor-in-chief of the JLA. The authors acknowledge BJET co-editor-in-chief Cathy Lewin, who joined the journal later in this process, and thus did not participate in authoring this piece.

The views expressed are those of the authors, and may not represent the views of the journals, scholarly societies, or other organisations to which they are affiliated.

The work was begun over the course of the lead author's sabbatical, which included periods co-located at UCL and KTH / the Swedish Digital Futures research centre, supporting direct interaction with editors based at those institutions.

## Author Contributions

**Conceptualization:** Simon Knight.

**Data curation:** Simon Knight.

**Formal analysis:** Simon Knight, Olga Viberg, Manolis Mavrikis, Vitomir Kovanović, Hassan Khosravi, Rebecca Ferguson, Linda Corrin.

**Investigation:** Simon Knight, Olga Viberg, Manolis Mavrikis, Vitomir Kovanović, Hassan Khosravi, Rebecca Ferguson, Linda Corrin.

**Methodology:** Simon Knight.

**Project administration:** Simon Knight.

**Validation:** Kate Thompson, Louis Major, Jason Lodge, Sara Hennessy, Mutlu Cukurova.

**Writing – original draft:** Simon Knight.

**Writing – review & editing:** Olga Viberg, Manolis Mavrikis, Vitomir Kovanović, Hassan Khosravi, Rebecca Ferguson, Linda Corrin, Kate Thompson, Louis Major, Jason Lodge, Sara Hennessy, Mutlu Cukurova.

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
