## [Decision Letter · Decision Letter 0]

10 Jun 2024

PONE-D-24-01120How do we learn about research ethics from published research? Developing editorial policy for emerging technologies using a scoping review and reference panelPLOS ONE

Dear Dr. Knight,

Thank you for submitting your manuscript to PLOS ONE. After careful consideration, we feel that it has merit but does not fully meet PLOS ONE’s publication criteria as it currently stands. Therefore, we invite you to submit a revised version of the manuscript that addresses the points raised during the review process.

We look forward to receiving your revised manuscript.

Kind regards,

Primus Che Chi, Ph.D

Academic Editor

PLOS ONE

Journal Requirements:

"The research reported was instigated as a collaboration between the Journal of Learning Analytics (JLA), Australasian Journal of Educational Technology (AJET), and the British Journal of Educational Technology (BJET), by the lead author, who was a previous co-editor-in-chief of the JLA. The authors acknowledge BJET co-editor-in-chief Cathy Lewin, who joined the journal later in this process, and thus did not participate in authoring this piece.

The views expressed are those of the authors, and may not represent the views of the journals, scholarly societies, or other organisations to which they are affiliated. 

The work was begun over the course of the lead author’s sabbatical, which included periods co-located at UCL and KTH / the Swedish Digital Futures research centre, supporting direct interaction with editors based at those institutions."

5. Please upload a copy of Figure 2, to which you refer in your text on page 38. If the figure is no longer to be included as part of the submission please remove all reference to it within the text.

6. Please ensure that you refer to Figure 2 and 3 in your text as, if accepted, production will need this reference to link the reader to the figure.

7. We note you have included a table to which you do not refer in the text of your manuscript. Please ensure that you refer to Table 5 in your text; if accepted, production will need this reference to link the reader to the Table.

8. We notice that your supplementary [figures/tables] are included in the manuscript file. Please remove them and upload them with the file type 'Supporting Information'. Please ensure that each Supporting Information file has a legend listed in the manuscript after the references list.

9. We note that this manuscript is a systematic review or meta-analysis; our author guidelines therefore require that you use PRISMA guidance to help improve reporting quality of this type of study. Please upload copies of the completed PRISMA checklist as Supporting Information with a file name “PRISMA checklist”.

**Additional Editor Comments:**

Dear Authors,

We have received some detailed and excellent feedback from two reviewers. The reviewers have raised substantial issues with the manuscripts and how these issued can be addressed in the next revision of the manuscript. The most important of the issues raised are the role of AI technology within the manuscript and coherence/connectedness of the various sections of the write-up. I am confident that if these issues are carefully reviewed and addressed by the authors it would strengthen the paper and improve the chances of being recommended for publication.

Best wishes!

Reviewers' comments:

Reviewer's Responses to Questions

**Comments to the Author**

1. Is the manuscript technically sound, and do the data support the conclusions?

Reviewer #1: Partly

Reviewer #2: Yes

2. Has the statistical analysis been performed appropriately and rigorously? 

Reviewer #1: N/A

Reviewer #2: I Don't Know

3. Have the authors made all data underlying the findings in their manuscript fully available?

Reviewer #1: Yes

Reviewer #2: Yes

4. Is the manuscript presented in an intelligible fashion and written in standard English?

Reviewer #1: Yes

Reviewer #2: Yes

5. Review Comments to the Author

Reviewer #1: Thank you for the opportunity to review “How do we learn about research ethics from published research? Developing editorial policy for emerging technologies using a scoping review and reference panel”. The paper seeks to develop a new editorial policy for research ethics across social science. The paper reports original research, not published elsewhere, though requires additional detail to sufficiently describe the research and resulting conclusions.

My comments focus on areas in which to provide more information to better align the stated goals of the paper with the actual achievements of the paper, and to sufficiently describe the research so as to understand what is done:

• The title does not align with the paper. I was expecting to see some published research heuristics, like work published by Abel Brodeur and others, but that wasn’t there and so I think the paper could be simply retitled to better address the actual content.

• The paper gets at an idea which is an important one: we are often missing clear guidelines with respect to ethical behavior in new technologies. AI has scrambled the academic world from top to bottom and we are all struggling to catch up. Understanding a new set of guidelines is essential. Unfortunately, the AI perspective of this feels completely missing or tacked on later to the paper. AI seems to be an addition, rather than the focus. I wonder if the framing could be revised so that AI is a single element, rather than pitched as the focus. Or could it be removed and added as a dynamic considered, just as a mentioned topic. Just reading as is, things feel unbalanced and undelivered, and so more detail is needed, or other reframing is required to better align the stated contribution with the actual research. This comment is the most significant in requiring revision before the paper could be published.

• Something that I inherently do not understand, with respect to editorial polices and research is: why can an executive decision not be made, and changes implemented? If a journal sets a new policy that X action or Y requirement must be implemented for acceptance, perhaps with a date in the future for implementation, why is this not an option? Some of this article feels fussy around the edge, without letting us understand, as readers, why editors cannot just define new, better, and perhaps more strict ethical criteria. I have long assumed it is inertia, so it would be useful to motivate why things are not simply changed.

• The conversation about de facto versus de jure use of IRB was very interesting. I appreciate the perspective that IRBs and similar review boards have been the focus of journals, although we all generally acknowledge that they are insufficient. I would have liked to see a bit more of a discussion of whether these requirements have achieved anything or if we should move away from them.

• I want to express some pushback on the recommendation for international IRB – or the perception that there should be a lack of it (on pages 3 and 4). It strikes me as close to soft bigotry. Many international organizations across the globe have excellent IRBs and many countries now have national offices. This recommendation feels very out of line with the current state of IRBs.

o The two references cited 17 and 18 are from 2002 and 2015. Things have significantly changed with respect to IRBs outside of “Western” contexts. The authors should either engage with this with respect to more recent publications; better contextualize that this comment is a perception of editors; and/or remove it.

• One of the strategy foci presented in what I believe to be Table 5 on page 29 is training. Is this a realistic recommendation. Several of the foci in Table 5 gave me similar pause, but this recommendation stood out. Given the extreme constraints facing journals and editors, is training an actually realistic recommendation?

• It would be useful to understand what defines social science. At present a lot of it seems to simply be “education”. Which is fine but is not encompassing of social science. If this is the case (e.g., that the focus is on education not necessarily social science), the entire paper should be reframed.

• Ultimately, I read this paper twice through my process of reading and reviewing. On my second reading, I came away with the impression that it was several disjointed sections adhered together. This, of course, is not necessarily relevant to PLOS ONE, which wants to publish unique and well-done work – which I do think this paper is. But, if this is to be a useful piece for anyone but the authors, significant revisions are needed to make it coherent, comprehensible, and actionable.

Reviewer #2: At the beginning the article foregrounds AI. I couldn't tell at that point if AI was to be taken as an example of an emerging technology that challenges research ethics in ways of concern in publication or if, more than a passing reference, AI was to the focus of the article and your methods. AI is not mentioned again until the end. It doesn't appear that concerns about AI in particular shaped the methodology, unless as a non-methodologist I have missed something. Therefore as a reader I would have found it helpful to have more clarity about AI as one among other emerging technologies, perhaps the most pressing on at the moment, that your approach would hope could be addressed by all the parties responsible for ethical research practices in published work. This appears to be merely a framing problem that should be easily resolved.

6. PLOS authors have the option to publish the peer review history of their article (what does this mean?). If published, this will include your full peer review and any attached files.

Reviewer #1: No

Reviewer #2: **Yes: **Jonathan D Moreno

---

## [Author Response · Author response to Decision Letter 0]

31 Jul 2024

Please see attached response to reviews

---

## [Decision Letter · Decision Letter 1]

19 Aug 2024

Emerging Technologies and Research Ethics: Developing Editorial Policy Using a Scoping Review and Reference Panel

PONE-D-24-01120R1

Dear Dr. Knight,

We’re pleased to inform you that your manuscript has been judged scientifically suitable for publication and will be formally accepted for publication once it meets all outstanding technical requirements.

Kind regards,

Primus Che Chi, Ph.D

Academic Editor

PLOS ONE

Additional Editor Comments (optional):

Dear Authors,

Thank you for carefully addressing the comments raised from the initial submission. After another round of review the reviewers have recommended that the manuscript should be accepted for publication, a recommendation that I agree to. Congratulations to the entire team for the amazing work that has gone into developing the manuscript to this stage.

Best wishes,

Primus

Reviewers' comments:

Reviewer's Responses to Questions

**Comments to the Author**

1. If the authors have adequately addressed your comments raised in a previous round of review and you feel that this manuscript is now acceptable for publication, you may indicate that here to bypass the “Comments to the Author” section, enter your conflict of interest statement in the “Confidential to Editor” section, and submit your "Accept" recommendation.

Reviewer #1: All comments have been addressed

Reviewer #2: All comments have been addressed

2. Is the manuscript technically sound, and do the data support the conclusions?

Reviewer #1: Yes

Reviewer #2: Yes

3. Has the statistical analysis been performed appropriately and rigorously? 

Reviewer #1: N/A

Reviewer #2: Yes

4. Have the authors made all data underlying the findings in their manuscript fully available?

Reviewer #1: Yes

Reviewer #2: Yes

5. Is the manuscript presented in an intelligible fashion and written in standard English?

Reviewer #1: Yes

Reviewer #2: Yes

6. Review Comments to the Author

Reviewer #1: Thank you for your thorough and thoughtful response. I appreciate you taking the time to address my remarks and to revise your manuscript.

Reviewer #2: I am satisfied with the responses to my concerns. The reframing of the manuscript meets the reservations I had about the original submission.

7. PLOS authors have the option to publish the peer review history of their article (what does this mean?). If published, this will include your full peer review and any attached files.

Reviewer #1: No

Reviewer #2: **Yes: **Jonathan D Moreno

---

## [Editor Report · Acceptance letter]

29 Aug 2024

PONE-D-24-01120R1 

PLOS ONE

Dear Dr. Knight, 

I'm pleased to inform you that your manuscript has been deemed suitable for publication in PLOS ONE. Congratulations! Your manuscript is now being handed over to our production team.

Kind regards, 

on behalf of

Dr. Primus Che Chi 

Academic Editor

PLOS ONE